# Effects of Lactobacillus Lactis Supplementation on Growth Performance, Hematological Parameters, Meat Quality and Intestinal Flora in Growing-Finishing Pigs

**DOI:** 10.3390/ani13071247

**Published:** 2023-04-04

**Authors:** Haitao Duan, Lizi Lu, Lei Zhang, Jun Li, Xu Gu, Junguo Li

**Affiliations:** 1College of Animal Science and Technology, Henan University of Animal Husbandry and Economy, Zhengzhou 450046, China; 2Feed Research Institute, Chinese Academy of Agricultural Sciences, Beijing 100081, China; 3Animal Husbandry Station of the Guangxi Zhuang Autonomous Region, Nanning 530221, China; 4Henan Institute of Agricultural, Animal and Aquatic Products Inspection & Testing Technology, Zhengzhou 450046, China

**Keywords:** growing-finishing pigs, *Lactobacillus lactis*, growth performance, meat quality, hematological parameters

## Abstract

**Simple Summary:**

Recently, many probiotics have been used as feed additives in the livestock industry for various purposes. Specifically, *Lactobacillus lactis* (LL) is effective as a growth promoter, anti-stress agent and immune enhancer. However, studies evaluating the effects of LL on growing-finishing pigs are limited. Therefore, in this study, we evaluated the growth performance of growing-finishing pigs using different doses of LL as a supplementary additive. The findings suggested that the inclusion of LL in the diet had a positive effect on growth performance in the grower phase and on meat quality.

**Abstract:**

Objective: The study was conducted to assess the effect of supplementation with *Lactobacillus lactis* (LL) on growth performance, hematological parameters, meat quality and intestinal flora in pigs from growing until slaughter. Methods: A total of 72 growing pigs (30.46 ± 3.08 kg) were randomly assigned to 3 groups (including 3 pens for each group, with 8 pigs in each pen). The three treatments comprised a basal diet (O-0) and two experimental diets supplemented for 14 weeks with 0.01% (O-100) and 0.03% (O-300) LL, respectively. Results: The final body weights of the pigs in the O-100 and O-300 groups were significantly higher (*p* < 0.05) than those of the O-0 group. In the grower phase, the average daily weight gain (ADG) and average daily feed intake (ADFI) of pigs fed the O-300 diet were higher (*p* < 0.05) than those of pigs fed the O-0 diet during the grower phase. BUN and MDA were significantly higher (*p* < 0.05 for all) in the O-0 group than in the O-100 and O-300 groups during the grower phase. No difference (*p* > 0.05) was observed in the hematological parameters among the three groups during the finisher phase. Counts of LL in the stomach were significantly higher (*p* < 0.05) in the O-300 group than in the O-0 group. Counts of *Escherichia coli* in the jejunum were significantly higher (*p* < 0.05) in the O-0 group than in the O-300 group. Furthermore, the hardness, cohesiveness, gumminess and resilience of longissimus dorsi muscle collected from pigs fed the O-300 diet were higher (*p* < 0.01; *p* = 0.024; *p* = 0.003; *p* = 0.014, respectively) than those of tissue collected from pigs fed the O-0 diet. Conclusion: Dietary LL supplementation increased final body weight, increased ADG in the grower phase and enhanced meat quality.

## 1. Introduction

Antibiotics have been widely used as growth promoters in animal feed since the 1950s. With the development of science and technology, there are more and more kinds of antibiotics, and they are more and more widely used. The abuse of antibiotics in the animal breeding industry has seriously threatened food safety and human health [1]. These health concerns led China to ban the use of AGPs in animal feed starting in 2020. The ban on adding antibiotics as growth promoters to animal feed has prompted scientists to develop alternative additives that could improve efficiency.

Probiotics have recently gained more interest, especially in swine production [2,3,4]. Pigs fed a compound probiotic diet had higher feed intake, daily weight gain, feed conversion ratio and lower incidence of diarrhea [4,5]. Likewise, supplementation with *L. plantarum* and *L. reuteri* complexes showed a reduction in levels of diarrhea in weaned piglets [6]. *Lactobacillus reuteri* strains have been studied as a therapeutic agent in acute diarrhea in young children [7]. Al-Rabadi et al. indicated that an efficient use of probiotics could be to improve feed utilization and animal health [8]. However, with a contradictory statement, Xu et al. reported that no significant effects were observed on growth performance in growing pigs fed bacillus-based probiotics [9]. To the best of our knowledge, the effects of probiotic supplements vary widely with diet composition, strain, animal age, dosage, and interactions with environmental factors [10,11]. Thus, our study was conducted to assess the effect of dietary LL on the growth performance, meat quality, hematological parameters and intestinal flora of grower and finisher pigs.

## 2. Materials and Methods

All experimental protocols were approved by the Animal Care and Use Committee of the Feed Research Institute of the Chinese Academy of Agricultural Sciences (ACE-CAAS-20180915), and the methods were in accordance with the relevant guidelines and regulations.

### 2.1. Animals, Diets and Experimental Design

Seventy-two castrated female pigs (Duroc × Landrace × Yorkshire) with a similar initial bodyweight (30.46 ± 3.08 kg) were randomly distributed into three treatment groups. There were three replicates (eight pigs per replicate) for each treatment.

The ingredients and nutrient contents of the experimental diets are given in Table 1. Two diets based on corn, soybean meal, cottonseed meal, and wheat bran were formulated to meet or exceed the recommendations of the Chinese National Feeding Standard for Swine (2004). Two diets were used for the grower phase (0 to 6 weeks) and the finisher phase (6 to 14 weeks). Macro materials were ground to a mean particle size of 535 μm through a 2.0 mm screen and mixed with vitamins and other ingredients in three separate batches before further processing. Three treatments were tested: (1) a basal diet without antibiotics and probiotics (control; O-0); (2) the basal diet supplemented with *Lactobacillus lactis* (LL, 5.0 × 10^12^ CFU/g) at 100 mg/kg (O-100); (3) the basal diet supplemented with LL at 300 mg/kg (O-300). All groups were pelleted (SZLH550X 170, Muyang, Yangzhou, China) to the same diameter (Table 2).

Pigs had free access to water and food. The same amount was made available twice, at 08:00 and 16:00. Each meal was weighed, the amount consumed was recorded and the daily intake was calculated. Exercise was canceled.

### 2.2. Growth Performance

The body weights were weighed at 0, 6 and 14 weeks to calculate the average daily gain (ADG) during 0–6 weeks, 6–14 weeks and 0–14 weeks. The feed intake was measured during 0–14 weeks to calculate the average daily feed intake (ADFI) and the gain to feed ratio (G:F) on a wet basis.

### 2.3. Hematological Parameters

Blood samples were collected on the final day in the grower phase, and finisher phase by Perceval vein puncture into a 10 mL pro-coagulation tube and an anticoagulative tube. Six samples were randomly collected from each pen. Serum and plasma were collected after centrifugation at 3000 rpm/min for 15 min at 4 °C and stored at −80 °C until analysis. The serum concentrations of cholesterol (CHO), glucose (GLU), high-density lipoprotein (HDL), low-density lipoprotein (LDL), malondialdehyde (MDA), IgM, IgG and IgA and the plasma concentrations of blood urea nitrogen (BUN) were determined using commercially available porcine ELISA kits (Nanjing Jiancheng Bioengineering Institute, Nanjing, China).

### 2.4. Meat Quality

One pig with medium weight per pen was chosen and slaughtered by exsanguination after electrical stunning. Samples (6 × 6 × 2.5 cm) of the longissimus dorsi muscle were immediately resected from the right side of the carcass and stored at 4 °C for 24 h before texture profile analysis (TPA) using a texture analyzer (XT2, Surrey, UK). TPA was carried out using an aluminum compression probe (5 mm diameter). The compression rate was set to 0.8 mm/s and the strain to 60%. Samples were compressed twice, 30 s apart. Six texture parameters, including chewiness, gumminess, cohesiveness, springiness, shear force and resilience, were calculated according to the methods described by Gines [12].

The color of fresh meat was measured using a Minolta chromameter (CR-10, Konica Minolta Sensing, Inc., Osaka, Japan) with 8 mm measuring diameter and 8° illumination angle (CIE standard illuminant D65). The CIE L * (light index), a * (red index) and b * (yellow index) were collected from three different orientations on the cut surface of each muscle chop. Calibration was carried out prior to each color determination using a white standard plate.

Further samples (2 × 3.5 × 5 cm) were cut from the center of the longissimus dorsi and weighed (W1). Each sample was hung in an inverted paper cup (250 mL volume) and stored at 4 °C for 24 h. Samples were then reweighed, after wiping off the surface diffusate (W2). Drip loss (DL) was calculated using the equation: DL (%) = [(W1 − W2)/W1] × 100.

### 2.5. Intestinal Flora

On the slaughter day of the study, fresh chylous samples were collected in the stomach, duodenum, jejunum, ileum of 3 pigs from each pen for enumerating the chylous microbial counts. Chylous and feed microbial load were assessed by using the pour plate method. The chylous sample and feed were mixed with normal saline (1:9 *w*/*v*), and after mixing, the supernatant was used for microbial counting. Specific agar plates were inoculated with 1 mL of appropriate dilution series (10^9^~10^12^) and incubated at 37 °C for 48 h. After incubation, the colonies were counted and expressed as CFU/g of chyme. MRS agar (Difco laboratory, Detroit, MI, USA) and Lysogeny broth agar (HiMedia, Einhausen, Germany) were used to count LL and *Escherichia coli*, respectively.

### 2.6. Statistical Analysis

Results were analyzed using one-way ANOVA. Multiple comparisons were carried out using Duncan’s multiple range test (SAS Inst. Inc., Cary, NC, USA). Covariance analysis was used to analyze growth performance in the finisher phase. The linear and quadratic effects of LL were assessed using regression analysis. Differences were considered statistically significant at *p* ≤ 0.05. Data were expressed as mean and pooled SEM.

## 3. Results

### 3.1. Growth Performance

In the grower phase, the initial body weight and G:F were not significantly different (*p* > 0.05 for all) between the groups receiving the different levels of LL (Table 3); however, the final body weight of pigs in the O-300 and O-100 groups was significantly higher (*p* < 0.05) than that of pigs in the O-0 group. The average daily gain (ADG) and average daily feed intake (ADFI) of pigs fed the O-300 diet was higher (*p* < 0.05) than that of pigs fed the O-0 diet. In the finishing phase, there were no significant effects (*p* > 0.05 for all) of the three different LL levels on final body weight, ADG, ADFI or G:F. In the full phase, there was no significant effects (*p* > 0.05 for all) of LL on ADG, ADFI or G:F.

### 3.2. Hematological Parameters

In the grower phase, BUN and MDA were significantly higher (*p* < 0.05 for all) in the O-0 group than in the O-100 and O-300 group (Table 4), and IgM was significantly higher (*p* < 0.05) in the O-0 group than in the O-300 group; however, there were no significant differences (*p* > 0.05) between O-0 group and O-100 group. In the finishing phase, no difference (*p* > 0.05) was observed in the hematological parameters among the three groups.

### 3.3. Meat Quality

There were no significant effects (*p* > 0.05 for all) of the different LL levels on drip loss, shear force, L *, a * or b * (Table 5). The hard, cohesiveness, gumminess and resilience of longissimus dorsi tissue collected from pigs fed the O-300 diet was higher (*p* < 0.05 for all) than that of tissue collected from pigs fed the O-0 diet (Table 5).

### 3.4. Intestinal Flora

There were no significant effects (*p* > 0.05 for all) of the different levels of LL on *Lactobacillus lactis* presence in the duodenum and jejunum or on the *Escherichia coli* microbial population in the stomach, duodenum, and ileum (Table 6). LL in the stomach was significantly higher (*p* < 0.05) in the O-300 group than in the O-0 group. Furthermore, LL in the ileum was significantly higher (*p* < 0.05) in the O-300 group than in the O-0 group. The *Escherichia coli* microbial population in the jejunum was significantly higher (*p* < 0.05) in the O-0 group than in the O-300 group.

## 4. Discussion

The benefits of dietary LL supplementation on growth performance, including G:F, antimicrobial activity, enzyme activity, and immune function, have been evidenced in several studies [1,13]. In a previous publication, Pupa et al. reported that Lactobacillus fermentum could improve the growth performance, carcass characteristics and muscular qualities of growing-finishing pigs [14]. In the present study, our results showed that dietary LL supplementation increased ADG and final body weight during the grower phase (Table 4). Increasing ADG could be ascribed to the fact that LL plays an important role in regulating digestive and absorptive functions [15]. This result agrees with Giang who reported that pigs fed with lactobacilli complex had higher ADG, ADFI and higher G:F [16]. Unexpectedly, no significant difference was observed in the current study during the 6–14 weeks trial period. The possible reasons for this discrepancy could be attributed to the intestinal microbiome environment. Initially, the intestinal flora of the growing pigs was inadequate and the effect of LL on growth performance was easier to observe. When the unstable intestinal condition has passed and a normal intestinal flora has been established, the LL effect will decline, which was similar to the previous research [17,18]. Furthermore, in the present study, BUN was significantly higher in the O-0 group than in the O-100 and O-300 group in the grower phase (Table 4). In the finishing phase, no difference was observed in BUN among three groups. BUN levels were inversely correlated with protein mass and absorption, suggesting that higher blood urea nitrogen levels indicate lower nitrogen absorption efficiency and increased lean body mass [19]. In addition, no significant differences were observed in LL populations in the duodenum and jejunum among groups under the conditions of our study, indicating that LL does not survive in the intestine of finishing pigs. Therefore, according to the results of our studies, the effects of LL may be related to growth stage of pigs. Evidence of probiotic gut mucosal colonization efficacy remains sparse and controversial; thus, the precise mechanism underlying this effect requires further investigations.

The color of the meat is important because it can affect consumer acceptance of the meat. Most consumers prefer red-pink pork to light-colored pork [16,20,21]. Choi noted that redness scores increased when prebiotics were added to the diet [22]. Zhang also consistently observed that the redness score of pigs in the probiotic-treated group was higher than in the control group [21]. No significant differences were observed in L *, a * and b * under conditions of our study, which contradicted previous research. Possible reasons for this difference could be nutritional factors (normal or reduced protein content) or the type of pigs used in the study (obese or lean genotype) [23,24,25,26,27]. Apart from meat color, cohesiveness, resilience and hardness are also considered to be the most important traits of meat quality. Previous investigations indicated the positive effect of direct-fed probiotic on meat quality [23,24]. In the present study, we assessed meat quality using TPA, which is based on people chewing food [12]. Pork from the O-300 and O-100 diet had a higher cohesiveness, hardness, gumminess and resilience value than from the O-0 diet (Table 5). Previously, researchers indicated that there was a strong relationship between muscle fibers type and the sensory traits of “cohesiveness” and “resilience” in pork [28,29,30,31]. The type of muscle fiber is determined by the expression of genes, such as *MyHC-I*, *MyHC-IIa* etc. [29]. Probiotics supplementation significantly promoted the expression of *MyHC-I* and *MyHC-IIa* and significantly decreased the mRNA abundance of *MyHC-Iix* [32,33]. At the protein level, probiotics significantly promoted the abundance of slow muscle protein in longissimus dorsi tissue, which indicated that pork from the probiotic diet had a better meat quality than from the control diet [33,34,35]. The results were consistent with our study.

## 5. Conclusions

Dietary LL supplementation increased final body weight and ADG in the grower phase, and improved hardness and chewiness of the longissimus dorsi. However, dietary supplementation with LL had no significant effects on the growth performance in the finisher phase. In summary, the results of this study suggest that LL can be used as a feed additive to enhance the growth performance of growing pigs and improve the meat quality of finishing pigs; however, whether it is suitable to use in the finisher phase or not still requires more investigation.

## Figures and Tables

**Table 1 animals-13-01247-t001:** Composition and nutrient levels of experimental diets (air-dry basis).

Items	Grower (0 to 6 Weeks)	Finisher (6 to 14 Weeks)
Ingredient (%, as-fed basis)		
Soybean meal	20.10	11.00
Corn	63.30	59.30
Wheat bran	6.00	15.77
Cottonseed meal	1.00	2.30
DDGS		7.50
Maize germ meal	6.00	
Mountain flour	1.00	1.20
Soybean oil	0.50	1.00
Calcium hydrophosphate	0.40	0.20
Salt	0.30	0.30
Calcium bicarbonate		0.10
L-lys	0.20	0.26
1% compound premix ^a^	1.00	1.00
L-Thr	0.13	0.07
DL-Met	0.07	
Chemical composition (%, as-fed basis) ^b^		
Digestible energy (kcal/kg)	3542	3442
Crude protein (%)	17.90	15.90
Calcium (%)	0.73	0.73
Total phosphorus (%)	0.55	0.45
Lys (%)	0.90	0.89
Met + Cys (%)	0.64	0.58

Note: ^a^ Premix provides following per kg of diet: Grower: vitamin A (retinyl acetate) 6 312 IU; vitamin D3 (cholecalciferol) 2 600 IU, vitamin E (DL-a-tocopheryl acetate) 35 IU; vitamin K3 (menadione sodium bisulphite) 4 mg; vitamin B1 (thiamin mononitrate) 2.8 mg; vitamin B2 (riboflavin) 5 mg; vitamin B6 (pyridoxine hydrochloride) 4 mg; vitamin B12 (cyanocobalamin) 28.1 μg; nicotinic acid 40 mg; folacin 1.1 mg; D-pantothenic acid 14 mg; biotin 44 μg; choline 400 mg; Cu (CuSO_4_·5H_2_O) 100 mg; Fe (FeSO_4_·H_2_O) 80 mg; Zn (ZnO) 75 mg; Mn (MnO) 40 mg; I (CaI_2_) 0.3 mg; Se (Na_2_SeO_3_) 0.3 mg. Finisher: vitamin A (retinyl acetate) 5612 IU; vitamin D3 (cholecalciferol) 2250 IU; vitamin E (DL-a-tocopheryl acetate) 21 IU; vitamin K3 (menadione sodium bisulphite) 4 mg; vitamin B1 2.8 mg; vitamin B2 5 mg; vitamin B6 (pyridoxine hydrochloride) 3 mg; vitamin B12 (cyanocobalamin) 21 μg; nicotinic acid 40 mg; folacin 1.1 mg; D-pantothenic acid 14 mg; biotin 62 μg; choline 360 mg; Cu (CuSO_4_·5H_2_O) 76 mg; Fe (FeSO_4_·H_2_O) 60 mg; Zn (ZnO) 50 mg; Mn (MnO) 20 mg; I (CaI_2_) 0.3 mg; Se (Na_2_SeO_3_) 0.3 mg. ^b^ CP was measured values, and others were calculated values.

**Table 2 animals-13-01247-t002:** The parameters of feed processing of the diets.

Item *	O-0	O-100	O-300
*Lactobacillus lactis* (5.0 × 10^12^ CFU/g), mg/kg	0	100	300
Conditioner before pelleting	MUTZ 600x2	MUTZ 600x2	MUTZ 600x2
Conditioning time, s	25	25	25
Temperature, °C	85	85	85
Pellets diameter, mm	3	3	3

* O-0, a basal diet without probiotics; O-100, a basal diet supplemented with *Lactobacillus lactis* at 100 mg/kg; O-300, a basal diet supplemented with *Lactobacillus lactis* at 300 mg/kg.

**Table 3 animals-13-01247-t003:** The effect of LL supplementation on the growth performance of growing and finishing pigs.

Item	O-0	O-100	O-300	SEM ^1^	*p*-Value ^2^		Overall *p*-Value
					Liner	Quadratic	
Grower phase (0 to 6 weeks)							
Initial body weight, kg	29.69	32.97	28.74	1.154	0.273	0.313	0.331
Final body weight, kg	49.99 ^a^	59.60 ^b^	57.17 ^b^	1.564	0.001	0.160	0.003
ADG, kg/d	0.48 ^a^	0.63 ^a,b^	0.68 ^b^	0.038	0.108	0.093	0.066
ADFI, kg	1.50 ^a^	1.83 ^a,b^	1.95 ^b^	0.088	0.140	0.102	0.086
G:F, kg/kg	0.32	0.35	0.35	0.008	0.241	0.447	0.388
Finisher phase (6 to 14 weeks)							
Initial body weight, kg	49.99 ^a^	59.60 ^b^	57.17 ^b^	1.564	0.001	0.160	0.003
Final bodyweight, kg	88.79	97.54	94.25	1.898	0.051	0.763	0.165
ADG, kg/d	0.69	0.68	0.66	0.016	0.735	0.585	0.807
ADFI, kg	2.03	2.16	2.06	0.034	0.146	0.635	0.336
G:F, kg/kg	0.34	0.32	0.31	0.007	0.176	0.725	0.403
Full phase (0 to 14 weeks)							
ADG, kg/d	0.6	0.66	0.67	0.018	0.225	0.339	0.310
ADFI, kg	1.81	2.02	2.01	0.052	0.100	0.342	0.176
G:F, kg/kg	0.33	0.33	0.33	0.004	0.485	0.849	0.784

Note: ^a,b^ Mean values within a row with different superscript letters are significantly different (*p* < 0.05); ADG, average daily gain; ADFI, average daily feed intake; G:F: gain: feed ratio. ^1^ Standard error of means. ^2^ Linear and quadratic effects of LL were evaluated using regression analysis.

**Table 4 animals-13-01247-t004:** The effect of LL supplementation on the hematological parameters of growing and finishing pigs.

Item	O-0	O-100	O-300	SEM	*p*-Value		Overall *p*-Value
					Liner	Quadratic	
Grower phase (0 to 6 weeks)							
Cholesterol, mmol/L	3.14	2.92	2.72	0.184	0.396	0.989	0.718
Glucose, mmol/L	2.48	3.76	4.33	0.396	0.470	0.628	0.144
BUN, mmol/L	3.31 ^b^	0.87 ^a^	0.79 ^a^	0.396	0.097	0.350	0.004
High density lipoprotein (HDL), mol/L	1.69	1.98	1.84	0.152	0.029	0.595	0.765
Low density lipoprotein (LDL), mmol/L	0.43	0.15	0.47	0.174	0.195	0.543	0.504
Malondialdehyde (MDA), nmol/mL	6.54 ^c^	5.53 ^b^	3.97 ^a^	0.310	0.268	0.004	0.000
IgM, g/L	0.07 ^b^	0.07 ^b^	0.05 ^a^	0.004	0.182	0.113	0.016
IgG, g/L	2.02	1.95	1.90	0.055	0.842	0.540	0.731
IgA, g/L	0.21	0.20	0.20	0.003	0.209	0.622	0.318
Finisher phase (6 to 14 weeks)							
Cholesterol, mmol/L	2.98	3.20	2.94	0.106	0.870	0.344	0.607
Blood sugar, mmol/L	2.48	3.76	4.33	0.396	0.470	0.628	0.144
BUN, mmol/L	0.69	0.75	0.67	0.030	0.202	0.057	0.582
High density lipoprotein (HDL), mol/L	2.46	2.54	2.70	0.136	0.634	0.174	0.790
Low density lipoprotein (LDL), mmol/L	0.61	0.27	0.51	0.175	0.739	0.242	0.389
Malondialdehyde (MDA), nmol/mL	7.52	5.31	5.13	0.661	0.077	0.390	0.273
IgM, g/L	0.07	0.09	0.07	0.009	0.082	0.539	0.346
IgG, g/L	1.83	1.82	1.86	0.067	0.701	0.801	0.972
IgA, g/L	0.20	0.20	0.21	0.003	0.729	0.657	0.370

Note: ^a,b,c^ Mean values within a row with different superscript letters are significantly different (*p* < 0.05).

**Table 5 animals-13-01247-t005:** The effect of LL supplementation on meat quality of finishing pigs.

Item at 3 d ^(1)^	O-0	O-100	O-300	SEM	*p*-Value		Overall *p*-Value
					Liner	Quadratic	
Drip loss, %	6.84	6.89	7.26	0.327	0.636	0.85	0.181
Shear force, N	34.09	34.37	26.71	2.443	0.241	0.470	0.805
L * (lightness) ^(2)^	69.45	68.89	69.9	0.235	0.471	0.130	0.368
a * (redness) ^(3)^	1.2	0.96	0.73	0.125	0.130	0.995	0.227
b * (yellowness) ^(4)^	−0.83	−1.18	−1.03	0.112	0.495	0.341	0.359
Hard, g	319.34 ^a^	1106.69 ^b^	1580.57 ^c^	168.286	0.014	0.21	0.031
Springiness, g	0.77	0.75	0.78	0.013	0.749	0.156	0.001
Cohesiveness, g	0.40 ^a^	0.46 ^ab^	0.48 ^b^	0.015	0.003	0.32	0.002
Gumminess, g	210.64 ^a^	510.00 ^b^	763.09 ^c^	83.586	0.011	0.236	0.003
Chewiness, g	569.49	389.08	594.82	68.228	0.009	0.209	0.567
Resilience, g	0.18 ^a^	0.18 ^a^	0.22 ^b^	0.008	0.000	0.799	0.014

Note: ^a,b,c^ Mean values within a row with different superscript letters are significantly different (*p* < 0.05); ^(1)^ All items were measured using longissimus dorsi stored at 4 °C for 3 days. ^(2)^ L * score varies from 0 (black) to 100 (white). ^(3)^ a * score varies from −60 (green) to +60 (red). ^(4)^ b * score varies from −60 (blue) to +60 (yellow).

**Table 6 animals-13-01247-t006:** Effect of different levels of LL on microbial population (log10 CFU/g fresh chyme) of pigs.

Item	O-0	O-100	O-300	SEM	*p*-Value		Overall *p*-Value
					Liner	Quadratic	
*Lactobacillus lactis*							
stomach	2.00 ^a^	5.14 ^b^	7.80 ^c^	0.841	0.000	0.26	0.000
duodenum	7.08	7.01	6.49	0.293	0.450	0.754	0.729
jejunum	7.76	7.31	7.27	0.176	0.291	0.616	0.522
ileum	7.66 ^a^	7.74 ^a,b^	8.21 ^b^	0.115	0.038	0.340	0.085
*Escherichia coli*							
stomach	1.43	1.00	1.00	0.145	0.244	0.506	0.422
duodenum	1.52	1.94	1.06	0.337	0.611	0.423	0.627
jejunum	5.21 ^b^	1.89 ^a^	1.00 ^a^	0.690	0.475	0.423	0.032
ileum	4.28	2.07	2.49	0.631	0.272	0.351	0.358

Note: ^a–c^ Mean values within a row with different superscript letters are significantly different (*p* < 0.05).

## Data Availability

The data presented in this study are available on request from the corresponding author. The data are not publicly available due to commercial reasons.

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
