# Peer review of "Effects of Lactobacillus Lactis Supplementation on Growth Performance, Hematological Parameters, Meat Quality and Intestinal Flora in Growing-Finishing Pigs"

_animals, 2023, doi:10.3390/ani13071247_

Round 1

Reviewer 1 Report (Previous Reviewer 2)

No comments 

Author Response

Best wishs for you!

Reviewer 2 Report (Previous Reviewer 1)

Overall, this version is a big improvement over the previous version. However, there are still some minor problems.

1. Please define the abbreviation correctly, for example, F:G: page 3 (2.2 section) defined as “feed:gain ratio” and in page 4 (table 3) defined as “feed consumption/ weight gain”.

2. The note section of table 3 - table 7, please change “P” as “p”.

3. “3.3 Meat quality” section, “The hard, cohesiveness, gumminess and resilience of longissimus dorsi tissue collected from pigs fed the O-300 diet was higher (p < 0.01; p = 0.024; p =0.003; p =0.014, respectively) than that of tissue collected from pigs fed the O-0 diet (Fig. 1).”, the p value is incorrectly marked. I think the p values marked in Figure 1 should be the whole p values of the three treatment groups, rather than the p values between the O-300 group and the O-0 group. Therefore, please re-analyze the difference between the two groups and correctly label the p value.

4. “Discussion” section, in the second paragraph, the author says that “Pork from the O-300 and O-100 diet had a higher cohesiveness, hard, gumminess and resilience value than from the O-0 diet (Fig.1, the larger area formed by index of group, the better meat quality).”, please add analysis between O-100 and O-0 in the “meat quality” section.

5. Some important references are recommended to be added, for example:

(1) Tang, X.; Liu, X.; Zhang, K. Effects of Microbial Fermented Feed on Serum Biochemical Profile, Carcass Traits, Meat Amino Acid and Fatty Acid Profile, and Gut Microbiome Composition of Finishing Pigs. Front. Vet. Sci. 2021, 8, 744630. doi: 10.3389/fvets.2021.744630

(2) Tang, X.; Liu, X.; Liu, H. Effects of Dietary Probiotic (Bacillus subtilis) Supplementation on Carcass Traits, Meat Quality, Amino Acid, and Fatty Acid Profile of Broiler Chickens. Front. Vet. Sci. 2021, 8, 767802. doi: 10.3389/fvets.2021.767802

Author Response

Overall, this version is a big improvement over the previous version. However, there are still some minor problems.

  1. Please define the abbreviation correctly, for example, F:G: page 3 (2.2 section) defined as “feed:gain ratio” and in page 4 (table 3) defined as “feed consumption/ weight gain”.

A: This has been corrected.

  1. The note section of table 3 - table 7, please change “P” as “p”.

A: This has been corrected.

  1. “3.3 Meat quality” section, “The hard, cohesiveness, gumminess and resilience of longissimus dorsi tissue collected from pigs fed the O-300 diet was higher (p < 0.01; p = 0.024; p =0.003; p =0.014, respectively) than that of tissue collected from pigs fed the O-0 diet (Fig. 1).”, the p value is incorrectly marked. I think the p values marked in Figure 1 should be the whole p values of the three treatment groups, rather than the p values between the O-300 group and the O-0 group. Therefore, please re-analyze the difference between the two groups and correctly label the p value.

A: Thank you for your valuable comments. The figure cannot better express meat quality, so I have corrected it to be a table.

  1. “Discussion” section, in the second paragraph, the author says that “Pork from the O-300 and O-100 diet had a higher cohesiveness, hard, gumminess and resilience value than from the O-0 diet (Fig.1, the larger area formed by index of group, the better meat quality).”, please add analysis between O-100 and O-0 in the “meat quality” section.

A: This has been corrected.

  1. Some important references are recommended to be added, for example:

(1) Tang, X.; Liu, X.; Zhang, K. Effects of Microbial Fermented Feed on Serum Biochemical Profile, Carcass Traits, Meat Amino Acid and Fatty Acid Profile, and Gut Microbiome Composition of Finishing Pigs. Front. Vet. Sci. 2021, 8, 744630. doi: 10.3389/fvets.2021.744630

(2) Tang, X.; Liu, X.; Liu, H. Effects of Dietary Probiotic (Bacillus subtilis) Supplementation on Carcass Traits, Meat Quality, Amino Acid, and Fatty Acid Profile of Broiler Chickens. Front. Vet. Sci. 2021, 8, 767802. doi: 10.3389/fvets.2021.767802

A: Some references have been added.

Reviewer 3 Report (Previous Reviewer 3)

The manuscript submitted for review, entitled Effects of Lactobacillus lactis supplementation on growth performance, haematological parameters, meat quality and intestinal flora in growing-finishing Pigs, concerns the evaluation of the effect of pig feed supplementation with probiotic bacteria on meat quality.

No over-representation of self-citation was found, and the experiment was generally carried out and described correctly.

However, in principle, the corrections made by the authors do not relate at all to my previous advice.

1. The authors still do not provide information on where and when the probiotic bacteria were administered, before or after the production of the feed.

2. The mechanism of action of the probiotic bacteria is still not explained - the authors only write that the mechanism is not known. Surely? However, there is some presumption otherwise why this research?

3. I still do not understand why in this case the colour of the meat was investigated (citing the reports of one other author is rather insufficient, especially as there were no differences between the groups).

4. The introduction does not explain the scope of the study and is laconic.

5. the discussion mostly ends with conclusions about the lack of research or contradictory results by the authors themselves, but with which results, where and when published? If an attempt is made to publish the results of exactly the same experiments and they are not repeated, the reliability of these studies remains questionable.

6. In many cases the results are not positive, i.e. there are no statistically significant differences. Are the studies therefore justified, or were the descriptors chosen incorrectly?

The overall assessment is that the article is written in a laconic manner and is not suitable for publication.

Author Response

The manuscript submitted for review, entitled Effects of Lactobacillus lactis supplementation on growth performance, haematological parameters, meat quality and intestinal flora in growing-finishing Pigs, concerns the evaluation of the effect of pig feed supplementation with probiotic bacteria on meat quality.

No over-representation of self-citation was found, and the experiment was generally carried out and described correctly.

However, in principle, the corrections made by the authors do not relate at all to my previous advice.

  1. The authors still do not provide information on where and when the probiotic bacteria were administered, before or after the production of the feed.

A: The treatments were tested: (1) a basal diet without antibiotics and probiotics (control; O-0); (2) the basal diet supplemented with Lactobacillus lactis (LL, 5.0×1012 CFU/g) at 100 mg/kg (O-100); (3) the basal diet supplemented with LL at 300 mg/kg (O-300).

  1. The mechanism of action of the probiotic bacteria is still not explained - the authors only write that the mechanism is not known. Surely? However, there is some presumption otherwise why this research?

A: Studies have shown that probiotics supplementation significantly promoted the  expression of MyHC-I and MyHC-Ⅱa and significantly decreased the mRNA abundance of MyHC-Ⅱx; at the protein level, probiotics significantly promoted the abundance of slow muscle protein in longissimus dorsi tissue, which indicated that pork from the probiotic diet had a better meat quality than from the control diet.

  1. I still do not understand why in this case the colour of the meat was investigated (citing the reports of one other author is rather insufficient, especially as there were no differences between the groups).

A: In the grower phase, the final body weight of the O-300 and O-100 group was significantly higher (p < 0.05) than that of the O-0 group, the average daily gain (ADG) and average daily feed intake (ADFI) of pigs fed O-300 diet was higher (p < 0.05) than that of pigs fed the O-0 diet. Although, there was no significant effects (p > 0.05 for all) of the three different LL levels on final body weight, ADG, ADFI or F:G in the finishing phase. Studies have shown that probiotics supplementation significantly promoted meat quality. So, I studied the effect of LL on meat quality.

  1. The introduction does not explain the scope of the study and is laconic.

A: This has been corrected.

  1. the discussion mostly ends with conclusions about the lack of research or contradictory results by the authors themselves, but with which results, where and when published? If an attempt is made to publish the results of exactly the same experiments and they are not repeated, the reliability of these studies remains questionable.

A: The positive effects of probiotics on meat quality have been confirmed by many studies.

  1. In many cases the results are not positive, i.e. there are no statistically significant differences. Are the studies therefore justified, or were the descriptors chosen incorrectly?

A: In the grower phase, the final body weight of the O-300 and O-100 group was significantly higher (p < 0.05) than that of the O-0 group, the average daily gain (ADG) and average daily feed intake (ADFI) of pigs fed O-300 diet was higher (p < 0.05) than that of pigs fed the O-0 diet.

Round 2

Reviewer 3 Report (Previous Reviewer 3)

no comments

Author Response

Best wishs for you!

This manuscript is a resubmission of an earlier submission. The following is a list of the peer review reports and author responses from that submission.

Round 1

Reviewer 1 Report

The present study investigated the effects of supplementation of lactic acid bacteria on the feed quality, growth performance, blood parameters, meat quality, and intestinal flora in pigs from growing until slaughter. This research is very meaningful, but the content is too complicated, and the author can't organize all the content well, which makes the logic of the paper very confusing.

Major comments

1. The summary of the research purpose is not accurate. In “Abstract” the authors say that “The objective of this study was to systematically determine the effects of supplementation of lactic acid bacteria on the feed quality, growth performance and blood parameters 6 in pigs from growing until slaughter”, while, in fact, meat quality, intestinal flora were also investigated.

2. “Materials and methods” are incomplete. Growth performance, meat color indicator (L*, a* or b*) were missing.

3. Did the authors consider the sex of the pigs at the time of sampling. Because the authors considered the sex of the pigs in the design of the experiment, which present in lines 87-88 that " ……distributed into three treatment groups, with eight pigs (4 sows and four barrows) per replicate, and three replicates per treatment." . 

4. line 89, “Barrows were provided with ad libitum access to water and feed.”, wrong description. The experimental animals included half sows and half barrows.

5. line 130, “Nine pigs were randomly selected from 9 groups and…”, wrong description.

6. line 176-179, whether dietary supplementation with 100 mg/kg (O-100) LAB has any effect on these indexes (cohesiveness, gumminess and resilience of longissimus dorsi) cannot be shown in figure 1.

7. line 254-256, “The content of MDA reflected redox status. However, there was no significant difference among the three groups in the finisher phase. Hence, the regulation of meat color by LAB needs further study.” What is the relationship between serum MDA index and meat color? This requires further discussion. Moreover, the authors did not measure MDA in longissimus dorsi.

8. line 262-267, the logic of these sentences is confused.

9. The data in this paper do not support the conclusions given by the authors. There was almost no discussion of the effect of the 100 mg/kg LAB throughout. I wonder why the authors think dietary LAB inclusion at 100 mg/kg could be appropriate in the present study.  

10. The manuscript has a lot of all kinds of errors, missing or wrong spaces, awkward wordings. It's hard to read. It is difficult to understand some sentences.

For examples:

Line 10, a space is need before “Results”;

Line 33, “Lan [6],” should change to “Lan et al. [6]”;

Line 36, “weaning pig`s [7].” ???

Line 39-40, references should add.

Line 77, “with Lactic Acid Bacteria (LAB, 5.0×1012 CFU/g) at”, Lactic Acid Bacteria was first appearing at line 48 and needs to be defined the first time it appears.

Line 137, “described by Gines.”, references need to be provided.

Line 181, “a,b Mean values within a row with different superscript letters are significantly different (P < 0.05).”, while “a,b” do not appear in this table.

Line 198, “Escherichia coli” should be italics.

Please note that this is just some of the examples, and there are many not listed. The most important thing is that some sentence is difficult to understand, so this paper needs to major revision of the language and format to be easy read.

Reviewer 2 Report

The study is a well-designed and well-written work. This study provides useful information on the effects of supplementation of lactic acid bacteria on the performance of pigs..

Comments to the authors

L8: .. for the group

L10: .. the final body weight……

L31: … performance, and nutrient digestibility…

L31: … Lactobacillus Plantarum and Lactobacillus reuteri

L51: add the approval number of the permission

L53: …. the Feed Laboratory Animal Welfare….

L73: … meal, and wheat bran….

L95: …. as the pellet durability index…

L100: was calculated according to Thomas and van der Poel (1996):

L119-120: ….Escherichia coli..

L182: …. All items were….

L187: … Blood parameters.. 

L193: …. High density lipoprotein…

L198: …. Escherichia coli

L202: ……. Escherichia coli

L228: … attention has been paid to… / …. in the recent…

L233: …. biotic supplementations…/ … Escherichia coli….

L235: …. indicates attention needs…

L241: …. higher, and the number of….

L257: ….. indicated the positive

L263: .. as a carbon source.. 

Reviewer 3 Report

The manuscript entitled Effects of lactic acid bacteria supplementation on growth per-formance and meat quality in growing-finishing Pigs submitted for review concerns the assessment of the effect of the addition of lactic acid bacteria with probiotic potential on selected growth parameters of pigs as well as chemical and qualitative features of selected tissues.

No self-citations were noticed in the manuscript and the selection of literature does not raise any objections.

In general, both the introduction and the description with the discussion of the results seem to be very sparse and do not sufficiently describe the topic.

In the opinion of the reviewer, the research was basically designed in accordance with the art of conducting such research, but there are several significant shortcomings:

1. It is not entirely true that there are not many reports on the impact of probiotic bacteria on the degree of feed utilization, health and quality of meat, including pigs. The studies described in this manuscript are not very innovative in this respect.

2. The authors used a specific bacterial preparation (5x10^12 cfu/g) at different doses. Why was the final level of these microorganisms not tested after the feed production process, where the temperature was above 80 degrees C? The dose used as a component is different from the actual amount of bacteria in the final feed. I consider this a big mistake.

3. It is quite obvious that the amount of LAB bacteria in the stomach increased with supplementation, but the effect of supplementation is not visible in the further sections of the digestive tract. You can also see the effect of reducing E. coli bacteria. What explains the lack of LAB growth and E. coli reduction? I am not convinced that this is the effect of the bacteria themselves, since they did not grow. Could it be the effect of bacteriocins?

4. It seems that some tests were performed without justification, because it could have been assumed in advance, for example, that the color of the meat would not change.

The authors have done a lot of work, however, taking into account the rank of the journal and the number of serious shortcomings in validity, form of experience and description, the article should be rejected.